# Redox Homeostasis, Gut Microbiota, and Epigenetics in Neurodegenerative Diseases: A Systematic Review

**DOI:** 10.3390/antiox13091062

**Published:** 2024-08-30

**Authors:** Constantin Munteanu, Anca Irina Galaction, Marius Turnea, Corneliu Dan Blendea, Mariana Rotariu, Mădălina Poștaru

**Affiliations:** 1Department of Biomedical Sciences, Faculty of Medical Bioengineering, University of Medicine and Pharmacy “Grigore T. Popa” Iasi, 700115 Iasi, Romania; anca.galaction@umfiasi.ro (A.I.G.); marius.turnea@umfiasi.ro (M.T.); mariana.rotariu@umfiasi.ro (M.R.);; 2Neuromuscular Rehabilitation Clinic Division, Clinical Emergency Hospital “Bagdasar-Arseni”, 041915 Bucharest, Romania; 3Department of Medical-Clinical Disciplines, General Surgery, Faculty of Medicine, “Titu Maiorescu” University of Bucharest, 0400511 Bucharest, Romania

**Keywords:** redox homeostasis, prooxidants, antioxidants, neurodegeneration, microbiota, epigenetics

## Abstract

Neurodegenerative diseases encompass a spectrum of disorders marked by the progressive degeneration of the structure and function of the nervous system. These conditions, including Parkinson’s disease (PD), Alzheimer’s disease (AD), Huntington’s disease (HD), Amyotrophic lateral sclerosis (ALS), and Multiple sclerosis (MS), often lead to severe cognitive and motor deficits. A critical component of neurodegenerative disease pathologies is the imbalance between pro-oxidant and antioxidant mechanisms, culminating in oxidative stress. The brain’s high oxygen consumption and lipid-rich environment make it particularly vulnerable to oxidative damage. Pro-oxidants such as reactive nitrogen species (RNS) and reactive oxygen species (ROS) are continuously generated during normal metabolism, counteracted by enzymatic and non-enzymatic antioxidant defenses. In neurodegenerative diseases, this balance is disrupted, leading to neuronal damage. This systematic review explores the roles of oxidative stress, gut microbiota, and epigenetic modifications in neurodegenerative diseases, aiming to elucidate the interplay between these factors and identify potential therapeutic strategies. We conducted a comprehensive search of articles published in 2024 across major databases, focusing on studies examining the relationships between redox homeostasis, gut microbiota, and epigenetic changes in neurodegeneration. A total of 161 studies were included, comprising clinical trials, observational studies, and experimental research. Our findings reveal that oxidative stress plays a central role in the pathogenesis of neurodegenerative diseases, with gut microbiota composition and epigenetic modifications significantly influencing redox balance. Specific bacterial taxa and epigenetic markers were identified as potential modulators of oxidative stress, suggesting novel avenues for therapeutic intervention. Moreover, recent evidence from human and animal studies supports the emerging concept of targeting redox homeostasis through microbiota and epigenetic therapies. Future research should focus on validating these targets in clinical settings and exploring the potential for personalized medicine strategies based on individual microbiota and epigenetic profiles.

## 1. Introduction

Neurodegenerative diseases (NDDs) represent a significant global health challenge [1], particularly in aging populations [2]. Alzheimer’s disease (AD) primarily affects individuals over 65 years old, with the risk doubling approximately every five years after this age threshold [3,4,5,6]. Parkinson’s disease (PD) affects about 1% of the population over 60 years old, with prevalence expected to rise dramatically due to increasing life expectancy [7,8,9,10]. By 2040, the number of individuals with PD is expected to become more than double, increasing the burden on healthcare systems and caregivers [11,12].

The pathophysiology of neurodegenerative diseases is complex, involving multiple molecular and cellular mechanisms [13], including mitochondrial dysfunction [14], oxidative stress [15,16], protein misfolding [17], excitotoxicity [18,19], and neuroinflammation [20]. 

Mitochondrial dysfunction leads to energy deficits and increased oxidative stress, contributing to neuronal damage. This is prominent in PD, AD, and HD [21]. Excessive ROS production and impaired antioxidant defenses result in oxidative damage to cellular components, exacerbating neurodegeneration [22]. The overactivation of glutamate receptors leads to increased intracellular calcium levels, triggering a cascade culminating in cell death, a mechanism relevant in ALS and AD [23,24]. The chronic activation of microglia and astrocytes releases pro-inflammatory cytokines, contributing to neuronal injury, a common feature in many neurodegenerative diseases [25,26,27,28,29].

Protein misfolding is a common feature in many neurodegenerative diseases [17,30]: AD is marked by beta-amyloid (Aβ) plaques and tau tangles [3,31], PD is marked by alpha-synuclein aggregates forming Lewy bodies [32], Huntington’s disease (HD) is marked by mutant huntingtin protein [33], and Amyotrophic lateral sclerosis (ALS) is marked by various protein aggregates like TAR DNA-binding protein 43 (TDP-43) and superoxide dismutase 1 (SOD1) [34].

Clinical manifestations of neurodegenerative diseases vary depending on the affected nervous system regions [35]. AD primarily affects cognitive function, leading to memory loss, language difficulties, disorientation, and behavior changes. Advanced stages result in severe dementia and functional impairment [36]. PD is characterized by motor symptoms like tremor, bradykinesia, rigidity, and postural instability, alongside non-motor symptoms like cognitive impairment, mood disorders, and autonomic dysfunction [37,38]. HD presents with motor, cognitive, and psychiatric symptoms, with chorea (involuntary movements) as a hallmark, along with cognitive decline and psychiatric disturbances like depression and irritability [39,40]. ALS affects motor neurons, leading to progressive muscle weakness, atrophy, and spasticity, with difficulty speaking, swallowing, and breathing as the disease progresses [41]. MS is characterized by central nervous system demyelination, leading to a wide range of neurological symptoms, including visual disturbances, muscle weakness, coordination problems, and cognitive deficits [42].

Current treatments for NDDs are largely symptomatic and aimed at improving quality of life, as there are no cures [43,44]. Pharmacotherapy for AD includes cholinesterase inhibitors (e.g., donepezil, rivastigmine) and NMDA receptor antagonists (e.g., memantine) to manage cognitive symptoms [45]. Recently, the FDA approved Eli Lilly’s Kisunla (formerly known as donanemab), an anti-amyloid drug for AD, marking a significant advancement in therapeutic strategies aimed at modifying the disease process rather than just alleviating symptoms [46]. PD treatments focus on dopaminergic therapies, including levodopa, dopamine agonists, and MAO-B inhibitors, with deep brain stimulation as an option for advanced cases [47]. In ALS, riluzole and edaravone may modestly slow disease progression [48]. Symptomatic treatments for spasticity include medications like baclofen and tizanidine, which are used in ALS and MS [49]. Psychiatric symptoms in diseases like HD and AD are managed with antidepressants, antipsychotics, and mood stabilizers [50,51,52]. Non-pharmacological interventions such as physical therapy help maintain mobility and reduce complications, occupational therapy assists patients in adapting to their environment and maintaining independence, and speech therapy addresses communication difficulties and swallowing problems [53,54]. Emerging therapies like gene therapy hold promise, particularly for monogenic disorders like HD and certain forms of ALS [55]. Immunotherapy targeting amyloid-beta and tau in AD and alpha-synuclein in PD is under investigation, while stem cell therapy aims to replace damaged neurons and support neural repair [56].

Recent research underscores the importance of the gut–brain axis, a bidirectional communication system linking the gastrointestinal tract and the central nervous system [57]. The gut microbiota, consisting of trillions of microorganisms, plays a crucial role in physiology [58]. Dysbiosis, or gut microbiota imbalance, is involved in the pathogenesis of NDDs [59]. An altered microbiota composition can influence oxidative stress levels in the brain by affecting the production of pro- and antioxidants [60,61]. Epigenetic modifications, including DNA methylation and histone modifications, also play critical roles in regulating oxidative stress responses [62,63]. These modifications can alter the expression of genes involved in antioxidant defenses, for example, by the hyper- or hypomethylation of genes encoding antioxidant enzymes, significantly impacting their expression and the cellular redox state [64,65,66]. Nutritional interventions, such as polyphenols, can induce beneficial epigenetic changes, enhancing antioxidant defenses and reducing inflammation [67,68,69].

This review examines the complex interplay between redox homeostasis, gut microbiota, and epigenetic modifications in the context of neurodegenerative diseases (NDDs). Redox homeostasis, the delicate balance between reactive oxygen species (ROS) and antioxidant defenses, is crucial for maintaining neuronal integrity. Disruptions in this balance, often exacerbated by dysbiosis—an imbalance in gut microbiota—lead to heightened oxidative stress, a key driver of neurodegeneration. The gut microbiota influences the host’s redox state by producing metabolites, such as short-chain fatty acids, which modulate oxidative stress and epigenetic mechanisms. Epigenetic modifications, including DNA methylation and histone modification, further regulate the expression of genes involved in oxidative stress responses and neuroinflammation. This intricate network of interactions suggests that therapeutic strategies targeting gut microbiota and epigenetic pathways may offer new avenues for restoring redox equilibrium and mitigating the progression of NDDs.

## 2. Methodology: PRISMA-Type Systematic Review Strategy

To achieve a comprehensive one-year systematic literature review on the role of oxidative stress, gut microbiota, and epigenetic modifications in neurodegenerative diseases and potential therapeutic strategies, we adhered to the principles outlined by the Preferred Reporting Items for Systematic Reviews and Meta-Analyses (PRISMA) [70] (Appendix A). The systematic review was conducted through a five-step PRISMA methodology (Figure 1).

Before commencing the standard systematic search, we conducted a preliminary, complex search to identify studies that specifically explored the interplay between redox homeostasis, oxidative stress, gut microbiota, epigenetics, and neurodegenerative diseases. This initial search was designed to capture studies that directly addressed the integrative nature of these mechanisms within titles and abstracts. We searched across four major international databases: PubMed, Scopus, Web of Science, and Google Scholar. The keywords and search strategies used were specifically tailored to identify studies that simultaneously discussed redox homeostasis, oxidative stress, gut microbiota, epigenetic modifications, and neurodegenerative diseases. This included the use of Boolean operators, positional terms, and truncations to refine the search. This initial search yielded a very limited number of studies—only four articles—that partially reflected the integrative focus we sought. These results highlighted the emerging and underexplored nature of the research area and emphasized the need to expand the scope of our review to include studies that, while not directly integrative, could contribute relevant insights when considered collectively.

Step 1: Database Interrogation. Following the preliminary search, we conducted a broader systematic search focusing on open-access articles published in 2024. This search aimed to capture the most recent developments in the field and included studies addressing individual aspects of oxidative stress, gut microbiota, and epigenetic modifications in neurodegenerative diseases. Open-access articles published in 2024 were searched on four international databases: PubMed, Scopus, Web of Science, and Google Scholar (Table 1).

Step 2: Initial Screening. The initial screening involved studies that met the inclusion criteria based on their titles and abstracts and were then subjected to a full-text review.

Step 3: Quality Assessment. The quality of the included studies was assessed using specific and focused keyword searches in the selected articles.

Step 4: Data Extraction. A standardized data extraction form was developed and used to extract relevant data from the included studies.

Step 5: Synthesis of Findings. The synthesis of findings was conducted in two parts:Narrative Synthesis: this part focused on the roles of oxidative stress, gut microbiota, and epigenetic modifications in neurodegenerative diseases.Quantitative Synthesis: this part focused on the microorganism’s composition of gut microbiota.

Articles identified through “other search” methods, including reference list checking and manual journal searches, were also used to ensure the comprehensiveness of the review and as-needed references to cover explanatory gaps.

## 3. Oxidative Stress and Redox Homeostasis in Neurodegenerative Diseases

Oxidative stress is increasingly recognized as a critical factor in the pathogenesis of neurodegenerative diseases (NDDs) [71]. It results from an imbalance between the production of reactive oxygen species (ROS) and reactive nitrogen species (RNS) and the capacity of antioxidant defenses [72]. Oxidative stress leads to the formation of advanced glycation end products (AGEs), which exacerbate disease progression through interactions with their receptors (RAGE). ROS and RNS, highly reactive molecules derived from oxygen and nitrogen, respectively, play essential roles as signaling molecules in normal physiological processes. However, their excessive production exacerbates oxidative stress [73].

Mitochondria are the primary source of ROS in cells, particularly during the electron transport chain’s activity in oxidative phosphorylation [74]. Neurons, with their high metabolic rate and dependence on mitochondrial function, are especially vulnerable to mitochondrial dysfunction. Mitochondrial dysfunction in neurons can disrupt ATP production, calcium homeostasis, and metabolic processes, leading to cell death and neurodegeneration [75]. This mitochondrial dysfunction is evident in various neurodegenerative conditions, where disrupted electron transport leads to excessive ROS generation [76].

Nicotinamide Adenine Dinucleotide Phosphate-oxidase (NADPH oxidase)—NOX is an enzyme complex that produces superoxide by transferring electrons from NADPH to oxygen. This enzyme has been implicated in excessive ROS production in many neurodegenerative diseases, further exacerbating oxidative stress and neuronal damage [77,78].

Nitric oxide synthase (NOS) produces nitric oxide (NO), a critical signaling molecule with numerous physiological functions. NO reacts rapidly with superoxide (O_2_−) to form peroxynitrite (ONOO−), a highly reactive nitrogen species (RNS). Peroxynitrite is a potent oxidant that induces significant oxidative and nitrative stress, contributing to cellular damage. It mediates the nitration of tyrosine residues in proteins, lipid peroxidation, and oxidative modifications to DNA, thereby exacerbating oxidative stress conditions commonly observed in neurodegenerative diseases (NDDs) [79,80].

In PD, mitochondrial complex I dysfunction and the oxidative metabolism of dopamine are pivotal factors contributing to the increased production of reactive oxygen species (ROS). The resultant oxidative stress exacerbates the degeneration of dopaminergic neurons, a hallmark of PD pathogenesis. This intricate interplay between mitochondrial dysfunction and dopamine oxidation underscores the central role of oxidative stress in PD. The cumulative effect of these oxidative insults not only accelerates the loss of dopaminergic neurons but also activates glial cells, perpetuating a chronic inflammatory response that further drives neuronal death [81,82,83].

In AD, the accumulation of amyloid-beta (Aβ) plaques leads to significant mitochondrial dysfunction, resulting in the increased production of reactive oxygen species (ROS). The presence of Aβ plaques disrupts the electron transport chain, particularly inhibiting cytochrome oxidase, which is crucial for cellular respiration and energy production. This disruption creates a feedback loop of oxidative damage and mitochondrial impairment, accelerating the progression of Alzheimer’s disease. Moreover, the inflammatory response elicited by Aβ plaques, characterized by the activation of microglia and astrocytes, further enhances ROS production and neuronal damage [84,85].

ALS is associated with superoxide dismutase 1 (SOD1) mutations, an enzyme that detoxifies superoxide radicals. Mutations in the SOD1 gene lead to a loss of function, accumulating superoxide radicals. This oxidative stress contributes significantly to motor neuron degeneration observed in ALS patients [86,87,88].

Huntington’s disease (HD) and Multiple Sclerosis (MS) also exhibit elevated ROS levels, with oxidative stress linked to mitochondrial dysfunction and energy metabolism deficits in HD [89] and inflammatory processes in MS [90].

**Antioxidant defenses**. These defenses are crucial for neutralizing excessive ROS/RNS and maintaining redox homeostasis, and are often compromised in neurodegenerative diseases, exacerbating oxidative stress and neuronal damage [91,92].

**Antioxidant Enzymes.** Superoxide dismutases (SODs) convert superoxide radicals into hydrogen peroxide, which are further detoxified by catalase and glutathione peroxidase (GPx) [93,94]. Mutations in SOD1 are directly linked to familial ALS, highlighting these enzymes’ importance in mitigating oxidative stress [95,96]. GPx reduces hydrogen peroxide and lipid peroxides using glutathione as a substrate. Reduced GPx activity has been observed in PD and AD, contributing to harmful peroxide accumulation [97]. Catalase breaks down hydrogen peroxide into water and oxygen, and its activity is often diminished in neurodegenerative conditions, further impairing the cell’s ability to detoxify ROS [98,99].

**Non-Enzymatic Antioxidants.** Glutathione (GSH) is a tripeptide and a major cellular antioxidant. The depletion of GSH is common in neurodegenerative diseases and correlates with increased oxidative damage. Reduced GSH levels in NDDs compromise the cell’s capacity to neutralize ROS and RNS, exacerbating oxidative stress [100]. Vitamins C and E act as free radical scavengers. Deficiencies or imbalances in these vitamins can increase neurons’ susceptibility to oxidative stress, further highlighting the importance of adequate antioxidant defenses [101].

**Interplay Between ROS/RNS and Antioxidant Defenses.** Maintaining a balance between ROS/RNS production and antioxidant defenses is crucial for cellular redox homeostasis. In neurodegenerative diseases, this balance is disrupted, leading to oxidative stress and subsequent neuronal damage [102]. ROS and RNS can induce lipid peroxidation, damaging cell membranes and disrupting cellular integrity. Lipid peroxidation impairs membrane-bound proteins and ion channels, contributing to neuronal dysfunction [103]. Oxidative modifications of proteins can lead to protein misfolding and aggregation, a hallmark of many neurodegenerative diseases [104]. For example, Aβ aggregation in AD [105] and α-synuclein aggregation in PD are driven by oxidative stress, contributing to toxic protein aggregates [106]. Additionally, oxidative stress can cause DNA damage, including strand breaks and base modifications, impairing neuronal function and viability and contributing to the progression of neurodegenerative diseases [107,108].

**Therapeutic Implications.** Strategies aimed at enhancing antioxidant defenses or reducing ROS/RNS production hold the potential to mitigate oxidative stress in NDDs. These strategies include using antioxidant supplements, enhancing endogenous antioxidant enzyme activity, and targeting sources of ROS/RNS production, such as mitochondrial support and NADPH oxidase inhibitors [109,110].

## 4. Gut Microbiota and Neurodegenerative Diseases

The intricate interplay between the gut and the brain, known as the gut–brain axis, has garnered significant attention for its role in maintaining overall health and potential implications in neurodegenerative diseases (NDDs). This complex, bidirectional communication system is mediated through neural, hormonal, and immunological pathways and influences brain function and homeostasis [111]. The gut–brain axis comprises a sophisticated network of communication channels linking the gastrointestinal tract and the central nervous system. This interaction is crucial for regulating various bodily functions and maintaining a stable internal environment. The vagus nerve transmits signals from the gut to the brain and vice versa, playing a pivotal role in controlling gastrointestinal motility, secretion, and neurotransmitter release, thereby directly influencing brain activity [112].

Gut-derived hormones like ghrelin, leptin, and insulin are essential players in this communication network. These hormones can cross the blood–brain barrier and interact with brain receptors, impacting brain function and behavior. Ghrelin and leptin regulate appetite and energy balance, while insulin influences cognitive functions [113,114].

Extracellular vesicles (EVs) from gut microbes, including gut microbiota-derived EVs (GMEVs), can transport proteins, lipids, and RNAs across the gut–brain axis, influencing neural cells and contributing to the modulation of neuroinflammatory and neurodegenerative processes. Due to their diverse cargo, these vesicles can affect various brain cells, including neurons, astrocytes, and microglia [115].

The gut microbiota, a vast community of trillions of microorganisms residing in the gastrointestinal tract (Table 2), is a key modulator of brain function [116]. These microorganisms produce neurotransmitters like serotonin [117], gamma-aminobutyric acid (GABA) [115], and dopamine [118], which are vital for brain health. The composition of the gut microbiota influences the production of various metabolites, including short-chain fatty acids (SCFAs), vitamins, and amino acids [119]. These metabolites affect brain function and behavior, highlighting the importance of a balanced gut microbiome [120].

The gut-associated lymphoid tissue (GALT) is integral to immune responses and inflammation. Immune molecules such as cytokines produced in the gut can travel to the brain, where they influence neuroinflammatory processes. This immunological pathway is crucial for understanding how gut health impacts brain health [121].

Gut bacteria produce a range of metabolites that play significant roles in maintaining brain health [122]. Short-chain fatty acids (SCFAs), in particular, have various beneficial effects on the brain, including reducing inflammation and promoting neurogenesis. SCFAs are produced by gut bacteria by fermenting dietary fibers [123]. Acetate, propionate, and butyrate are the most studied SCFAs, each with distinct and beneficial effects on brain health. SCFAs can cross the blood–brain barrier and exert neuroprotective effects by reducing inflammation, enhancing barrier integrity, and promoting neurogenesis. Butyrate, in particular, helps maintain the integrity of the blood–brain barrier, protecting the brain from toxins and pathogens. Moreover, SCFAs have been implicated in promoting neurogenesis and improving synaptic plasticity, which is crucial for learning and memory. SCFAs can influence the production and release of neurotransmitters, affecting mood and cognitive functions, underscoring the significant impact of gut health on mental health and cognitive performance [124].

The gut microbiota helps maintain the integrity of the intestinal barrier, preventing pathogens and toxins from translocating and triggering systemic inflammation. A compromised gut barrier can lead to increased intestinal permeability, or “leaky gut”, allowing harmful substances to enter the bloodstream and affect brain health [113].

Dysbiosis, an imbalance in the gut microbiota composition, has been implicated in developing neurodegenerative diseases [118]. This imbalance can influence brain health through various mechanisms, including the modulation of oxidative stress levels in the brain. Dysbiosis can increase intestinal permeability, allowing lipopolysaccharides (LPSs) and other pro-inflammatory molecules to enter the bloodstream [112]. These substances can trigger systemic inflammation and oxidative stress in the brain, contributing to the pathogenesis of NDDs. Changes in the gut microbiota can alter immune responses, leading to chronic neuroinflammation [125], a common feature in neurodegenerative diseases such as Alzheimer’s and Parkinson’s diseases. Dysbiosis can affect the production of oxidative metabolites, increasing ROS/RNS levels and impairing antioxidant defenses in the brain, a critical factor in neuronal damage and disease progression [60].

## 5. Therapeutic Strategies Targeting Gut Microbiota

Recent advancements in understanding the intricate roles of gut microbiota have paved the way for innovative therapeutic strategies targeting these aspects.

**Probiotics**: Probiotics are beneficial bacteria live microorganisms that, administered in adequate amounts, confer health benefits, maintain gut microbiota balance, produce neuroactive compounds like GABA and serotonin, and reduce inflammation by regulating cytokines, thereby improving symptoms of anxiety, depression, and cognitive decline [126]. Specific strains such as Lactobacillus and Bifidobacterium have shown promise in influencing brain health. These probiotics can alter gut microbiota composition, enhance barrier integrity, and reduce inflammation, impacting neurodegenerative processes [127].

**Prebiotics**: Prebiotics are non-digestible food ingredients that selectively stimulate the growth and activity of beneficial gut bacteria. Compounds like inulin and fructooligosaccharides enhance the production of short-chain fatty acids (SCFAs), such as butyrate, which possess neuroprotective properties [128]. By fostering a favorable gut environment, prebiotics reduce oxidative stress and improve brain function [60,118,129].

**Synbiotics**: Synbiotics refer to a combination of probiotics and prebiotics that act synergistically to confer health benefits to the host. In the context of NDDs, synbiotics are particularly relevant due to their potential to modulate the gut–brain axis [130].

**Postbiotics**: Postbiotics are bioactive compounds produced by probiotics (beneficial bacteria) during fermentation. These compounds include various metabolites, such as SCFAs, peptides, proteins, polysaccharides, vitamins, and other bioactive molecules. Unlike probiotics, which are live microorganisms, postbiotics are non-living components that still confer health benefits to the host [131].

**Fecal Microbiota Transplantation (FMT)**: FMT involves transplanting fecal bacteria from a healthy donor to a patient, aiming to restore a healthy gut microbiota composition. This procedure has shown potential in treating various conditions, including NDDs, by reducing gut dysbiosis and inflammation. However, FMT must be approached with caution due to the risks of infection and the necessity for rigorous donor screening [60,116,121,129].

**Dietary Interventions**: The Mediterranean diet, rich in fruits, vegetables, whole grains, and healthy fats, has been associated with reduced inflammation and oxidative stress. This dietary pattern supports gut health and has potential benefits for individuals with NDDs by modulating gut microbiota and reducing systemic inflammation. The ketogenic diet [132], characterized by high fat and low carbohydrate intake, induces ketosis, producing ketones that provide neuroprotective effects. This diet reduces oxidative stress and improves mitochondrial function, offering potential benefits for neurodegenerative conditions [68,123,125,133].

**Table 2 antioxidants-13-01062-t002:** Quantitative synthesis of the microorganism composition of gut microbiota, focusing on its modulation and impact on neurodegenerative diseases—data from the selected articles.

Microorganism Phylum	Relative Abundance (%) in Healthy Individuals	Role in Gut Microbiota	Changes in Neurodegenerative Diseases	Microorganism Species	Number of Mentions in the Selected Articles	Refs. no.
**Bacteroidetes**	30	Metabolism of complex molecules, production of SCFAs (anti-inflammatory properties)	Decrease in abundance, linked to reduced anti-inflammatory properties	Bacteroides	6	[104,118,127,129]
				Prevotella	4	[104,129]
**Firmicutes**	50	The phylum Firmicutes is one of the major bacterial groups present in the human gut microbiota. Fermentation of dietary fibers, production of SCFAs (gut barrier integrity, immune response modulation)	Variable, but often altered ratio with Bacteroidetes, linked to various diseases	Firmicutes	6	[104,118,127]
				Lactobacillus	158	[104,112,117,118,121,125,127]
				Clostridium	5	[104,117,127]
				Bacillus	1	[118]
				Enterococcus	2	[118]
				Streptococcus	12	[118]
				Ruminococcus	4	[118]
				Eubacterium	4	[104,118,129]
**Proteobacteria**	10	Includes many pathogenic bacteria, can contribute to inflammation and oxidative stress	Increase in abundance, associated with heightened inflammation and oxidative stress	Escherichia	6	[104,118]
				Helicobacter	1	[104]
**Actinobacteria**	5	Includes beneficial bacteria such as Bifidobacterium (gut health maintenance, vitamin production)	Decrease in abundance, linked to increased gut permeability and systemic inflammation	Bifidobacterium	59	[104,112,118,127]
**Verrucomicrobia**	2	Includes Akkermansia muciniphila (mucin degradation, improved metabolic health, reduced inflammation)	Decrease in abundance, related to metabolic health and inflammation	Akkermansia muciniphila	4	[104,127]

## 6. Hormesis, Nutrients (Polyphenols and Probiotics) in Gut–Brain Axis Disorders

Hormesis in nutritional therapies leverages the adaptive response to moderate stress, enhancing resilience against higher stress levels, crucial for managing gut–brain axis disorders through polyphenols and probiotics. Polyphenols, found in various fruits and vegetables, activate the Nrf2 pathway, upregulating antioxidant enzymes, reducing oxidative stress, and modulating gut microbiota by promoting beneficial bacteria like Bifidobacterium and Lactobacillus. Polyphenols can cross the blood–brain barrier, offering direct neuroprotective effects, such as reducing amyloid-beta aggregation in Alzheimer’s disease models. The combination of polyphenols and probiotics provides synergistic effects, enhancing bioavailability, comprehensively modulating the gut–brain axis, and offering neuroprotective and cognitive benefits. This combination can improve cognitive functions, reduce symptoms of mental health disorders, and slow neurodegenerative disease progression by maintaining a healthy gut–brain axis. Thus, leveraging the principles of hormesis with polyphenols and probiotics presents a promising approach to enhancing cellular resilience, maintaining gut microbiota balance, and protecting against neurodegenerative processes, offering holistic strategies for brain health and overall well-being [134,135].

## 7. Epigenetic Modifications and Oxidative Stress in Neurodegenerative Diseases

Epigenetic modifications, heritable changes in gene expression without alterations in the DNA sequence, encompass mechanisms such as DNA methylation, histone modifications, and non-coding RNAs [136]. These changes are pivotal in regulating oxidative stress responses and maintaining cellular homeostasis [137]. In NDDs, aberrant epigenetic modifications can exacerbate oxidative stress, contributing significantly to disease pathogenesis [138]. Understanding the interplay between epigenetic modifications and oxidative stress opens potential therapeutic avenues for treating neurodegenerative diseases [139].

DNA methylation involves adding a methyl group to cytosine residues in DNA, typically at CpG dinucleotides [140]. This modification can suppress gene expression by preventing the binding of transcription factors or recruiting repressive protein complexes [141].

The hypermethylation of antioxidant genes such as superoxide dismutase (SOD) and glutathione peroxidase (GPx) can reduce their expression, leading to decreased cellular defense against oxidative stress. Conversely, the hypomethylation of pro-oxidant genes can increase their expression, contributing to higher ROS production and oxidative damage [142].

In Alzheimer’s disease (AD), altered DNA methylation patterns have been observed in genes involved in amyloid precursor protein processing and tau phosphorylation. These changes contribute to oxidative stress and neurodegeneration [143]. Similarly, in Parkinson’s disease (PD), differential methylation in genes related to dopamine metabolism and mitochondrial function links epigenetic changes to oxidative stress and neuronal death [144], pressing gene expression. By inhibiting these enzymes, DNA methylation inhibitors can reactivate silenced neuroprotective genes [145]. DNA methylation inhibitors can restore the expression of neuroprotective genes and reduce oxidative stress by reversing aberrant DNA methylation patterns, which is particularly promising for conditions where the hypermethylation of critical genes exacerbates disease progression [146]. Compounds like resveratrol and curcumin can modulate epigenetic marks and enhance antioxidant defenses, providing neuroprotection in models of neurodegenerative diseases. These polyphenols, found in various dietary sources, influence DNA methylation and histone modifications, mitigating oxidative stress [147].

Histones are proteins around which DNA is wrapped to form nucleosomes. Post-translational modifications of histones, such as acetylation, methylation, phosphorylation, and ubiquitination, can influence chromatin structure and gene expression. The acetylation of histones by histone acetyltransferases (HATs) generally promotes gene transcription by relaxing chromatin structure. Deacetylation by histone deacetylases (HDACs) condenses chromatin, leading to transcriptional repression [148].

Histone modifications can regulate the expression of genes involved in the oxidative stress response. For instance, the acetylation of histones at the promoters of antioxidant genes can enhance their expression and improve cellular resistance to oxidative damage [149]. In Huntington’s disease (HD), disrupted histone acetylation leads to the decreased expression of neuroprotective genes and increased oxidative stress [141]. In Amyotrophic lateral sclerosis (ALS), HDAC inhibitors have demonstrated the potential to reduce oxidative stress and improve motor neuron survival [150].

Histone Deacetylase (HDAC) Inhibitors: HDAC inhibitors prevent the removal of acetyl groups from histone proteins, leading to a more relaxed chromatin structure and increased gene transcription. By enhancing the expression of neuroprotective genes and antioxidant defenses, HDAC inhibitors can mitigate oxidative stress. Drugs like valproic acid and suberoylanilide hydroxamic acid (SAHA) have shown potential in preclinical models of NDDs, reducing oxidative damage and improving neuronal survival. HDAC inhibitors can enhance the expression of antioxidant genes and protect against oxidative damage. HDAC inhibitors are being investigated for their potential in treating neurodegenerative diseases such as AD and HD. These drugs can reactivate silenced genes and restore antioxidant defenses by modulating histone acetylation.

Non-coding RNAs, including microRNAs (miRNAs) [151] and long non-coding RNAs (lncRNAs), are involved in the post-transcriptional regulation of gene expression [144]. These molecules can influence oxidative stress by targeting mRNAs encoding proteins involved in ROS production and antioxidant defenses.

MiRNAs can downregulate the expression of antioxidant enzymes. Modulating miRNAs offers a strategy to influence oxidative stress and inflammation pathways. For example, miR-21 targets SOD1, increasing neurons’ oxidative stress. The dysregulation of miRNAs has been linked to various neurodegenerative diseases, where they modulate oxidative stress pathways and contribute to disease progression [152].

LncRNAs can act as molecular sponges for miRNAs, modulating their availability and activity. By sequestering miRNAs that target antioxidant genes, lncRNAs can indirectly influence oxidative stress responses. LncRNAs have been implicated in regulating genes involved in mitochondrial function and oxidative stress, playing significant roles in neurodegenerative diseases [143,153].

**Nutritional Epigenetics**: Polyphenols, such as resveratrol, curcumin, and green tea polyphenols, can modulate epigenetic marks, enhancing antioxidant defenses and reducing inflammation [154]. These compounds, found in various dietary sources, influence DNA methylation and histone modifications, mitigating oxidative stress and offering neuroprotective benefits. S-adenosylmethionine (SAM), a key methyl donor in DNA methylation processes, has been explored to counteract epigenetic dysregulation in NDDs. Supplementation with SAM can influence gene expression and potentially ameliorate disease progression by restoring proper methylation patterns [155].

## 8. The Interplay between Redox Homeostasis, Gut Microbiota, and Epigenetics in Neurodegenerative Diseases

The intricate interplay between redox homeostasis, gut microbiota, and epigenetics (Figure 2) is critical in understanding the pathogenesis of neurodegenerative diseases such as Alzheimer’s, Parkinson’s, and Huntington’s diseases. Redox homeostasis refers to the delicate balance between producing reactive oxygen species (ROS) and the antioxidant defenses that neutralize them. The disruption of this balance leads to oxidative stress, which causes significant damage to cellular components, including lipids, proteins, and DNA, which are particularly detrimental in neurons due to their high metabolic rate and limited regenerative capacity [156].

Gut microbiota, the diverse community of microorganisms residing in the human gastrointestinal tract, significantly influences redox homeostasis. These microbes produce various metabolites, including short-chain fatty acids (SCFAs) like butyrate, propionate, and acetate, which have profound effects on the host’s immune system and oxidative stress levels. SCFAs, particularly butyrate, are known for their antioxidant properties, which help scavenge ROS and enhance the host’s antioxidant defenses, thereby protecting neurons from oxidative damage [159].

Epigenetic modifications, including DNA methylation, histone modification, and non-coding RNA expression, are deeply intertwined with redox status and gut microbiota composition. These modifications regulate gene expression patterns related to inflammation, oxidative stress, and neuronal survival. For example, oxidative stress can lead to changes in DNA methylation, which may activate or suppress genes involved in neuroinflammation and synaptic plasticity, both of which are critical in neurodegenerative disease progression [160].

Gut microbiota can directly influence epigenetic modifications by producing SCFAs, which act as histone deacetylase (HDAC) inhibitors. This inhibition leads to increased histone acetylation, a modification associated with activating genes involved in anti-inflammatory responses and neuroprotection. Additionally, microbial metabolites can modulate the expression of microRNAs, which in turn regulate gene expression at the post-transcriptional level, further linking gut microbiota to the epigenetic regulation of neurodegenerative processes [161].

Dysbiosis, or the imbalance of gut microbiota, has been linked to increased oxidative stress and aberrant epigenetic modifications in the brain. This imbalance can result from various factors, including diet, antibiotic use, and environmental stressors, leading to reduced beneficial bacteria and increased pathogenic species. The resulting dysbiosis exacerbates oxidative stress by decreasing the production of SCFAs and other beneficial metabolites, thereby weakening the gut–brain axis and promoting neurodegeneration [162].

Furthermore, the interaction between gut microbiota and the immune system is crucial in modulating neuroinflammation, a key driver of neurodegenerative diseases. Microbial metabolites like SCFAs can modulate the activity of regulatory T cells (Tregs), which are essential for maintaining immune tolerance and preventing chronic inflammation in the brain. Dysbiosis can disrupt this balance, leading to chronic neuroinflammation and subsequent neuronal death, which are characteristic of diseases like Alzheimer’s and Parkinson’s [163].

Gasotransmitters such as nitric oxide (NO), hydrogen sulfide (H_2_S), and carbon monoxide (CO) are crucial integrators of redox homeostasis, gut microbiota, and epigenetic regulation, playing significant roles in the pathogenesis of neurodegenerative diseases [164]. These gaseous molecules act as signaling agents that influence oxidative stress responses, inflammatory pathways, and gene expression, forming a complex network that links the gut–brain axis with epigenetic modifications [165]. NO modulates vascular tone and neurotransmission but can exacerbate oxidative stress when dysregulated. It influences gene expression through the S-nitrosylation of proteins involved in chromatin remodeling and DNA methylation. The gut microbiota modulates NO levels, further linking it to the gut–brain axis. H_2_S, known for its potent antioxidant properties, enhances the activity of antioxidant enzymes and promotes the growth of beneficial gut bacteria, which produce neuroprotective metabolites like butyrate. H_2_S also influences epigenetic processes by modulating histone acetylation and DNA methylation, impacting genes related to oxidative stress and inflammation [139]. CO, produced during heme degradation, exerts anti-inflammatory effects and helps maintain gut microbiota balance by regulating the immune response and preserving intestinal barrier integrity. It also modulates histone modifications, influencing the expression of genes crucial for redox homeostasis. Together, these gasotransmitters form a critical link between redox balance, gut microbiota composition, and epigenetic regulation, highlighting their potential as therapeutic targets for neurodegenerative diseases by addressing the interconnected pathways that drive disease progression [166].

Therapeutic interventions targeting the gut microbiota offer promising avenues for the prevention and treatment of neurodegenerative diseases. Probiotics, prebiotics, and dietary interventions aimed at restoring a healthy gut microbiome can enhance the production of beneficial metabolites like SCFAs, thereby improving redox balance and modulating epigenetic regulation in the brain. For example, diets rich in fiber can promote butyrate-producing bacteria growth, enhancing the neurons’ antioxidant defenses [163].

Epigenetic therapies, which aim to reverse aberrant epigenetic modifications, also hold potential in treating neurodegenerative diseases. By targeting specific epigenetic enzymes such as HDACs and DNA methyltransferases, these therapies can restore normal gene expression patterns, reducing neuroinflammation and oxidative stress. Integrating gut microbiota-targeted therapies with epigenetic interventions could provide a synergistic approach to modulating the key pathways involved in neurodegeneration [167].

## 9. Limitations of the Study

This systematic review examines the latest research in neurodegenerative diseases, oxidative stress, gut microbiota, and epigenetics by restricting the analysis to studies published in 2024. While this approach ensures that the review captures cutting-edge developments, it inherently excludes significant research from previous years, potentially limiting the context and depth of the findings. The scarcity of studies that specifically address the interplay between oxidative stress, gut microbiota, and epigenetics further constrains the comprehensiveness of the review, as only a few articles partially reflecting these connections were identified. Expanding the temporal scope to include studies from the past 5 to 10 years could have provided a more robust and integrated understanding, but this would have resulted in an unmanageable volume of articles, leading to challenges in data synthesis and potential redundancy. Consequently, the decision to focus on the year 2024 was made to balance relevance and practicality, though this may narrow the generalizability of the conclusions. The insights provided by this review should be interpreted as reflective of recent advances rather than a comprehensive overview of the field.

## 10. Conclusions and Future Directions

This review underscores the significant roles of oxidative stress, gut microbiota, and epigenetic modifications in the pathogenesis of neurodegenerative diseases. While these factors have been individually associated with disease progression, their interconnectedness suggests a more complex interaction that warrants further investigation. Understanding how oxidative stress influences gut microbiota composition and function, or how epigenetic changes may modulate oxidative stress responses and microbial profiles, could reveal novel insights into disease mechanisms. Further studies should focus on identifying and validating specific oxidative stress markers, such as ROS levels and antioxidant enzyme activities, which have been consistently linked to disease progression. Additionally, research should explore how oxidative stress may affect gut microbiota and whether changes in microbiota composition can, in turn, influence oxidative stress, potentially creating a feedback loop that exacerbates neurodegenerative processes. Similarly, research should target particular gut microbiota taxa, like *Bifidobacterium* and *Lactobacillus*, which were frequently associated with neuroprotective effects. Specific epigenetic modifications, including DNA methylation patterns and histone acetylation levels, should be further explored as potential diagnostic markers or therapeutic targets. Building on the evidence, clinical trials are needed to evaluate the efficacy of combining antioxidant therapies with microbiota-targeted interventions (e.g., probiotics) and epigenetic drugs, such as HDAC inhibitors. For example, a trial investigating the synergistic effects of polyphenol-rich diets, *Bifidobacterium* supplementation, and HDAC inhibitors could provide insights into effective combination therapies. These trials should be designed to measure short-term neuroprotective outcomes and long-term impacts on disease progression and patient quality of life. A more comprehensive exploration of the interplay between oxidative stress, gut microbiota, and epigenetic modifications could uncover novel therapeutic strategies and lead to a more holistic approach to the treatment of neurodegenerative diseases.

## Figures and Tables

**Figure 1 antioxidants-13-01062-f001:**
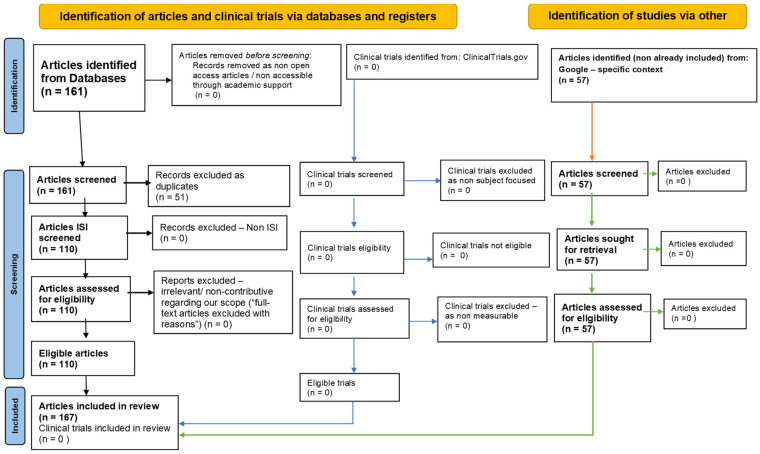
The PRISMA flow diagram is used to illustrate the flow of information process [70].

**Figure 2 antioxidants-13-01062-f002:**
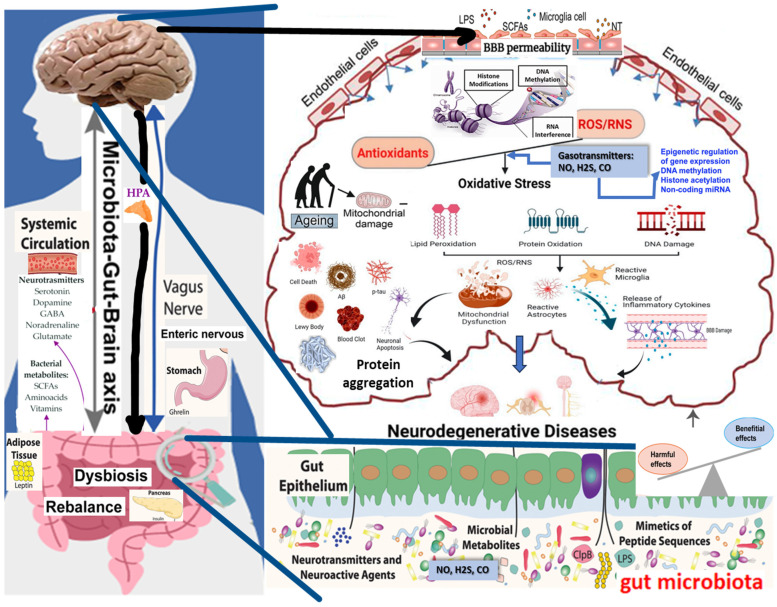
The interplay between diet, gut microbiota, epigenetic regulation, and oxidative stress in neurodegenerative diseases. The image depicts the gut–brain axis. It highlights the impact of diet, nutrients, and environmental agents on intestinal microbiota, shows oxidative stress pathways, and indicates the disruption of the blood–brain barrier (BBB). The figure highlights the role of gut-derived metabolites in modulating epigenetic mechanisms, such as DNA methylation and histone modifications. These epigenetic changes influence the expression of genes involved in inflammatory and oxidative stress responses. Furthermore, the figure depicts how dysbiosis, an imbalance in gut microbiota, can disrupt these protective mechanisms, leading to increased oxidative stress and aberrant epigenetic modifications. The figure was inspired by [157,158].

**Table 1 antioxidants-13-01062-t001:** The specific keyword combinations used for searching relevant scientific articles.

Keywords	PubMed	Scopus	Web of Science	Google Scholar	Total
“Neurodegenerative Disease” + “Oxidative Stress”	7	1	0	0	8
“Alzheimer’s Disease” + “Oxidative Stress”	13	3	14	22	53
“Parkinson’s Disease” + “Oxidative Stress”	8	8	9	14	39
“Huntington’s Disease” + “Oxidative Stress”	1	1	1	2	5
“Amyotrophic Lateral Sclerosis” + “Oxidative Stress”	4	0	0	3	7
“Multiple Sclerosis” + “Oxidative Stress”	6	1	5	9	21
“Gut Microbiota” + “Neurodegenerative Disease”	2	0	3	3	8
“Microbiota” + “Neurodegenerative Disease”	4	0	4	3	11
“Epigenetic” + “Neurodegenerative Disease”	1	1	0	0	2
“DNA Methylation” + “Neurodegenerative Disease”	0	0	0	0	0
“Histone Modification” + “Neurodegenerative Disease”	0	1	0	0	1
“RNA” + “Neurodegenerative Disease”	2	1	1	1	5

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
