# Peer review of "Redox Homeostasis, Gut Microbiota, and Epigenetics in Neurodegenerative Diseases: A Systematic Review"

_antioxidants, 2024, doi:10.3390/antiox13091062_

Round 1

Reviewer 1 Report

The aim of the paper is to explore the balance between prooxidants and antioxidants in neurodegenerative diseases, examining the roles of gut microbiota and epigenetic modifications in modulating oxidative stress. The review highlights the complex interplay between oxidative stress and neurodegeneration, emphasizing the potential therapeutic strategies targeting redox homeostasis, gut microbiota, and epigenetic pathways. Strengths of the paper include a comprehensive literature review, integration of multiple biological mechanisms, and the identification of novel therapeutic avenues.

The most notable limitation of this systematic review is its restriction to studies published exclusively in the year 2024. This temporal limitation potentially omits significant research and advancements made in previous years, which are crucial for a comprehensive understanding of the topic. I recommend expanding the review period to include studies from at least the past 5 to 10 years. This broader temporal scope would provide a more robust and contextualized foundation for the conclusions drawn and would ensure that the review captures the continuity and evolution of scientific knowledge in the field of neurodegenerative diseases, oxidative stress, gut microbiota, and epigenetics. Addressing this issue would enhance the validity and relevance of your findings.

The title is generally descriptive, but it could be more concise. Consider "Redox Homeostasis, Microbiota, and Epigenetics in Neurodegenerative Diseases: A Systematic Review".

The abstract provides a broad overview but lacks specific details about methodologies, key findings, and implications. It should be more precise in summarizing the key points.

The introduction effectively sets the context for the review but is too verbose. Consider streamlining to focus on the most critical points.

Results are presented, but the tables (like Table 2 on microbiota composition) need clearer headings and explanations to improve readability.

Ensure all figures and tables are self-explanatory with appropriate legends and captions.

The discussion broadly interprets the findings but often repeats the introduction. Focus more on integrating the results with existing literature and identifying gaps.

The limitations section is comprehensive but could be condensed. Focus on the most significant limitations and how they impact the study’s conclusions.

The conclusion effectively summarizes the review but should emphasize actionable future research directions.

This section is speculative. Provide more concrete suggestions based on the review findings.

Author Response

Reviewer 1

Major comments

The aim of the paper is to explore the balance between prooxidants and antioxidants in neurodegenerative diseases, examining the roles of gut microbiota and epigenetic modifications in modulating oxidative stress. The review highlights the complex interplay between oxidative stress and neurodegeneration, emphasizing the potential therapeutic strategies targeting redox homeostasis, gut microbiota, and epigenetic pathways. Strengths of the paper include a comprehensive literature review, integration of multiple biological mechanisms, and the identification of novel therapeutic avenues.

Thank you for your positive summary and thoughtful evaluation of our manuscript, which reflects the central aim of our paper—to explore the balance between prooxidants and antioxidants in neurodegenerative diseases, with a focus on how gut microbiota and epigenetic modifications influence oxidative stress and contribute to neurodegeneration. We appreciate your acknowledgment of the complex interplay we have highlighted between these factors and the potential therapeutic strategies we have proposed.

Detail comments

The most notable limitation of this systematic review is its restriction to studies published exclusively in the year 2024. This temporal limitation potentially omits significant research and advancements made in previous years, which are crucial for a comprehensive understanding of the topic. I recommend expanding the review period to include studies from at least the past 5 to 10 years. This broader temporal scope would provide a more robust and contextualized foundation for the conclusions drawn and would ensure that the review captures the continuity and evolution of scientific knowledge in the field of neurodegenerative diseases, oxidative stress, gut microbiota, and epigenetics. Addressing this issue would enhance the validity and relevance of your findings.

We appreciate the comments concerning the temporal limitation of our systematic review, particularly the concern that restricting the review to studies published exclusively in 2024 may omit significant research and advancements from previous years. Our decision to focus on studies published in 2024 was driven by the objective of capturing the most recent developments and the latest trends in research. We aimed to highlight the current state of knowledge and emerging concepts that have yet to be thoroughly integrated into earlier reviews. In considering the reviewer's recommendation, we conducted an initial exploration to determine the feasibility of expanding the temporal scope. Notably, very few studies specifically capture the intricate interplay between oxidative stress, gut microbiota, and epigenetics in neurodegenerative diseases. Our search across databases yielded only four articles that partially reflect this connection. This scarcity of studies underscores the emerging nature of this research area and the limited integration of these topics. On the other hand, extending the search period for the previously performed review would result in an overwhelmingly large number of articles, making it impractical to manage within a short and optimal timeframe. Additionally, the redundancy of findings from earlier years compared to those published in 2024 would diminish the utility of this approach. The significant increase in research interest in these topics has led to a substantial volume of publications in 2024, which capture our study's most relevant and up-to-date data. Given these considerations, we believe that maintaining the temporal focus in 2024 will provide a targeted and efficient approach to synthesizing the most pertinent information while avoiding the pitfalls of information overload and redundancy.

The title is generally descriptive, but it could be more concise. Consider "Redox Homeostasis, Microbiota, and Epigenetics in Neurodegenerative Diseases: A Systematic Review".

We appreciate the reviewer's suggestion to refine the title of our manuscript for conciseness. The proposed title, "Redox Homeostasis, Microbiota, and Epigenetics in Neurodegenerative Diseases: A Systematic Review," captures our study's key elements more succinctly. Therefore, we will adopt the proposed title: "Redox Homeostasis, Microbiota, and Epigenetics in Neurodegenerative Diseases: A Systematic Review."

The abstract provides a broad overview but lacks specific details about methodologies, key findings, and implications. It should be more precise in summarizing the key points.

We appreciate the reviewer's comments on the abstract and recognize the need to provide a more precise summary that includes specific details about our systematic review's methodologies, key findings, and implications.

The introduction effectively sets the context for the review but is too verbose. Consider streamlining to focus on the most critical points.

We appreciate the reviewer's feedback on the introduction and acknowledge the concern that it may be too verbose. We have carefully reviewed the current introduction and removed redundant information not directly contributing to framing the research question or the review's objectives. On the other hand, we have added extra information to respect all reviewers' comments.

Results are presented, but the tables (like Table 2 on microbiota composition) need clearer headings and explanations to improve readability.

We appreciate the reviewer's attention to the presentation of the results, particularly the observation that the tables, such as Table 2 on microbiota composition, would benefit from clearer headings and more detailed explanations to improve readability. We will revise the headings in Table 2 to ensure that they are more descriptive and informative.

Ensure all figures and tables are self-explanatory with appropriate legends and captions.

We appreciate the reviewer's emphasis on the importance of ensuring that all figures and tables are self-explanatory and have appropriate legends and captions. We have revised the legends and captions of all figures and tables to ensure they provide sufficient context and explanation.

The discussion broadly interprets the findings but often repeats the introduction. Focus more on integrating the results with existing literature and identifying gaps.

While broadly interpreting the findings, we appreciate the reviewer's observation that the discussion section tends to repeat content from the introduction rather than focusing on integrating the results with existing literature and identifying research gaps. We have improved the discussion, interpreting the findings in the context of the results obtained.

The limitations section is comprehensive but could be condensed. Focus on the most significant limitations and how they impact the study’s conclusions.

Thank you for your feedback on the limitations section. We agree that a more focused approach would be beneficial. We will condense the limitations section to emphasize the most significant issues: the restriction to studies published in 2024, which may limit the comprehensiveness of our findings; the scarcity of studies addressing the interplay between oxidative stress, gut microbiota, and epigenetics; and the challenges of managing a broader temporal scope. We have revised the section accordingly.

The conclusion effectively summarizes the review but should emphasize actionable future research directions. This section is speculative. Provide more concrete suggestions based on the review findings.

Thank you for your feedback on the conclusion. We agree that providing more concrete suggestions based on the review findings would strengthen this section. In response, we have revised the conclusion to focus on specific, actionable recommendations drawn directly from the evidence reviewed.

Reviewer 2 Report

The source of the adapted figures needs to be indicated. For example, Figs 1 and 4 seem to resemble too much the ones in other publications.

The FDA has just approved Eli Lilly’s Kisunla, formerly known as donanemab, an anti-amyloid drug for AD. It would be worth mentioning in the Introduction part.

N/A

Author Response

Reviewer 2

Major comments

The source of the adapted figures needs to be indicated. For example, Figs 1 and 4 seem to resemble too much the ones in other publications.

Thank you for bringing this important issue to our attention. We acknowledge the necessity of properly attributing the source of any figures adapted from other publications. The included figures are original but inspired by preexisting data used in the review process.

The FDA has just approved Eli Lilly’s Kisunla, formerly known as donanemab, an anti-amyloid drug for AD. It would be worth mentioning in the Introduction part.

Thank you for highlighting the recent FDA approval of Eli Lilly’s Kisunla (donanemab), an anti-amyloid drug for Alzheimer's disease (AD). We agree that this significant development is highly relevant to the context of our review and should be mentioned in the Introduction. We have revised the Introduction to include a brief discussion of Kisunla’s approval. This addition will serve to underscore the ongoing advancements in therapeutic strategies for neurodegenerative disease.

Reviewer 3 Report

The title of the manuscript suggests that it is a review exploring the role of microbiota and epigenetics in the redox balance of neurodegenerative diseases, which is a very relevant topic in the research about neurodegeneration. However, the manuscript shows a compilation of data about redox balance, microbiota, and epigenetics in neurodegenerative diseases without clearly establishing how these topics are interrelated; it needs more critical analysis expected for a systematic review. The above is also caused by how the authors selected the articles to be reviewed since the Keywords used were not combined in such a way that articles were selected in which the selected topics were discussed. The authors are recommended to rethink how to combine the keywords to carry out a more targeted search on the subject they wish to develop; they can use various operators (for example, Boolean, positional, truncations). Due to the above, the publication of this manuscript is only recommended, with first undergoing a significant restructuring based on a critical and in-depth analysis of the search terms and, subsequently, of the selected literature. Also, authors must revise more original papers than review papers. 

* The introduction could be more specific, guiding the reader to the research question and resolving any potential doubts in advance. Instead of highlighting the work's importance, the introduction is lost in providing data irrelevant to the review's topic. For instance, treatments against neurodegenerative diseases that are not discussed later. The definition of oxidative stress is placed at the end of the introduction, rather than as a starting point to discuss prooxidants and antioxidants. The introduction should explain the methodological strategy followed, such as why only articles published in 2024 were analyzed, why only open access, and why reviews but not original articles. As a systematic review, it explores the data reported in the literature. Therefore, the statement that “This review explores the intricate balance between pro- and antioxidants, elucidating the mechanisms contributing to oxidative stress in NDDs and discussing potential therapeutic strategies targeting redox homeostasis” is imprecise.

* Table 1 contains 160 articles, which need to be consistent with the initial 161 articles reported in Figure 1. Therefore, it is recommended that the consistency of the data be verified.

* Explaining or describing how articles classified as “other search” were searched is desirable.

* In the figure captions, the meaning of the abbreviations used in the figure should be included.

* In Figure 2, another way to place the information on the left could be useful to enlarge the rest of the image since the font size is too small and cannot be read. In addition, it is recommended not to include terms not used or explained in the rest of the work, such as “synbiotics” or “postbiotics”.

Author Response

Reviewer 3

Major comments

The title of the manuscript suggests that it is a review exploring the role of microbiota and epigenetics in the redox balance of neurodegenerative diseases, which is a very relevant topic in the research about neurodegeneration. However, the manuscript shows a compilation of data about redox balance, microbiota, and epigenetics in neurodegenerative diseases without clearly establishing how these topics are interrelated; it needs more critical analysis expected for a systematic review. The above is also caused by how the authors selected the articles to be reviewed since the Keywords used were not combined in such a way that articles were selected in which the selected topics were discussed. The authors are recommended to rethink how to combine the keywords to carry out a more targeted search on the subject they wish to develop; they can use various operators (for example, Boolean, positional, truncations). Due to the above, the publication of this manuscript is only recommended, with first undergoing a significant restructuring based on a critical and in-depth analysis of the search terms and, subsequently, of the selected literature. Also, authors must revise more original papers than review papers. 

Thank you for your thoughtful and detailed feedback on our manuscript. We greatly appreciate your insights and recognize the importance of addressing your concerns to enhance the quality and impact of our review. We have carefully considered your comments in the context of the manuscript's focus and methodology, and we outline below the steps we will take to address these issues comprehensively.

Comment: The restriction to studies published exclusively in 2024 may limit the comprehensiveness of the review by omitting significant research and advancements from previous years.

  • Response: We understand that a broader temporal scope could provide a more comprehensive understanding of the evolution and continuity of scientific knowledge in the fields of neurodegenerative diseases, oxidative stress, gut microbiota, and epigenetics. Our decision to focus on studies published in 2024 was driven by the objective of capturing the most recent developments and emerging trends in these rapidly evolving fields. We aimed to highlight new concepts and findings that have not been fully integrated into earlier reviews. However, we recognize that this approach may limit the context and depth that could be gained from a broader temporal analysis. During our initial exploration of expanding the temporal scope, we found that very few studies specifically capture the intricate interplay between oxidative stress, gut microbiota, and epigenetics in neurodegenerative diseases, with only four articles partially reflecting this connection. Extending the search period would likely result in an overwhelming number of articles, which could introduce redundancy and make the review impractical to manage within a short and optimal timeframe. Given these considerations, we believe that maintaining the temporal focus in 2024 will allow us to efficiently synthesize the most pertinent and up-to-date information while avoiding the pitfalls of information overload. We will clearly communicate this rationale and address its implications in the limitations section to ensure transparency and understanding.

Comment: The manuscript compiles data on redox balance, microbiota, and epigenetics in neurodegenerative diseases but lacks a clear connection between these topics and needs more critical analysis.

  • Response: We acknowledge the need for a more cohesive and critically analyzed presentation of how oxidative stress, gut microbiota, and epigenetic modifications interact in the context of neurodegenerative diseases. To address this, we have restructured the discussion section to better integrate these topics, emphasizing their interconnections and discussing their combined impact on disease progression.

Comment: The current selection of articles may not effectively target studies that discuss the interrelated aspects of the topics due to the way keywords were combined. A more targeted search using various operators is recommended.

  • Response: After conducting this targeted search, we found that very few studies capture the intricate connections between these key topics. Specifically, our search yielded only four articles that partially reflect this integrative focus. This scarcity highlights the emerging nature of this research area and the limited integration of these topics in the existing literature.

Comment: The manuscript relies heavily on review articles rather than original research.

  • Response: We recognize the value of including more original research to strengthen the empirical foundation of our review. The aforementioned limitations explain this situation and allow us to cover the review subject efficiently.

Comment: The introduction should explain the methodological strategy, including the rationale for focusing on studies published in 2024, why only open-access articles were included, and why the emphasis was on review articles rather than original research.

  • Response: We have explained our methodological choices to ensure clarity and transparency. We will clearly articulate the reasons for limiting the review to studies published in 2024, focusing on capturing the latest research and emerging trends. We have also explained why we focused on open-access articles, which were intended to ensure accessibility and reproducibility, and why review articles were initially prioritized for their broad overview of the current state of knowledge. This explanation will help readers understand the scope and limitations of our review and the choices made to ensure it remains current, accessible, and relevant.

Comment: The title suggests a focus on the interrelation between microbiota, epigenetics, and redox balance, which was not fully realized in the manuscript.

  • Response: Given the planned restructuring to integrate these topics better, we believe the new title will reflect the manuscript's content more accurately after revisions. However, we will ensure that the title accurately conveys the scope and focus of the revised manuscript, potentially refining it to better align with the enhanced critical analysis and integration of the key topics.

Detail comments

* The introduction could be more specific, guiding the reader to the research question and resolving any potential doubts in advance. Instead of highlighting the work's importance, the introduction is lost in providing data irrelevant to the review's topic. For instance, treatments against neurodegenerative diseases that are not discussed later. The definition of oxidative stress is placed at the end of the introduction rather than as a starting point to discuss prooxidants and antioxidants. The introduction should explain the methodological strategy followed, such as why only articles published in 2024 were analyzed, why only open access, and why reviews but not original articles. As a systematic review, it explores the data reported in the literature. Therefore, the statement that “This review explores the intricate balance between pro- and antioxidants, elucidating the mechanisms contributing to oxidative stress in NDDs and discussing potential therapeutic strategies targeting redox homeostasis” is imprecise.

Thank you for your insightful comments on our manuscript's introduction. We recognize the importance of ensuring that the introduction is clear, focused, and directly aligned with our review's research question and objectives.

Comment: The introduction should be more specific, clearly guiding the reader to the research question and resolving potential doubts in advance.

  • Response: We have revised the introduction to ensure that it clearly and succinctly presents the research question early.

Comment: The introduction currently includes irrelevant data and places the definition of oxidative stress at the end rather than using it to introduce key concepts.

  • Response: We revised the introduction. The definition and role of oxidative stress will be moved to the beginning of the introduction to set the foundation for the subsequent discussion on prooxidants, antioxidants, gut microbiota, and epigenetics.

Comment: The introduction should explain the methodological strategy, including why only articles published in 2024 were analyzed, why only open-access articles were included, and why reviews rather than original articles were emphasized.

  • Response: To address this, we have the methodology section. We have explained that the decision to focus on articles published in 2024 was made to capture the most recent developments and trends in the field. We will also clarify that we prioritized open-access articles to ensure the accessibility and reproducibility of our review.

Comment: The statement that “This review explores the intricate balance between pro- and antioxidants, elucidating the mechanisms contributing to oxidative stress in NDDs and discussing potential therapeutic strategies targeting redox homeostasis,” is imprecise.

  • Response: We have revised the focus of our review. The revised statement will emphasize that the review synthesizes recent findings related to oxidative stress, gut microbiota, and epigenetics in neurodegenerative diseases, focusing on understanding how these factors interact and influence disease progression.

* Table 1 contains 160 articles, which need to be consistent with the initial 161 articles reported in Figure 1. Therefore, it is recommended that the consistency of the data be verified.

Thank you for identifying the discrepancy between the number of articles reported in Table 1 and Figure 1. As a result of the first stage of review, we have made the necessary corrections in the new version, considering also the newly used articles.

* Explaining or describing how articles classified as “other search” were searched is desirable.

Thank you for your insightful comment regarding the need for clarity on how articles classified as "other search" were identified and included in our review. The "other search" category in our review refers to articles that were identified through methods outside the primary database searches. This includes articles found through reference list checking and manual searches of key journals. These methods were employed to capture any relevant studies that might have been missed by the primary search strategy, particularly those that are highly relevant but may not have appeared in the initial database queries due to variations in indexing or keyword usage.

* In the figure captions, the meaning of the abbreviations used in the figure should be included.

Thank you for your important observation regarding using abbreviations in our figure captions. We agree that it is essential to ensure that all abbreviations are clearly defined within the figure captions to enhance clarity and accessibility for all readers.

* In Figure 2, another way to place the information on the left could be useful to enlarge the rest of the image since the font size is too small and cannot be read. In addition, it is recommended not to include terms not used or explained in the rest of the work, such as “synbiotics” or “postbiotics”.

Thank you for your valuable feedback on Figure 2. We understand the importance of ensuring that all figures are easily readable and that the terminology used is consistent with the manuscript's content. We will update the caption for Figure 2 to reflect any changes made to the figure and to ensure that the figure is fully self-explanatory, including the definitions of any abbreviations or terms that remain in the figure.

Round 2

Reviewer 1 Report

The manuscript has been sufficiently improved to warrant publication in Antioxidants.

The manuscript has been sufficiently improved to warrant publication in Antioxidants.

Author Response

Thank you very much for your positive feedback and for recognizing the improvements made to our manuscript.

Reviewer 3 Report

The paper's development has improved substantially since the first revision. However, the conclusion's wording can still be improved. It can suggest not only that oxidative stress, gut microbiota, and epigenetic modifications are relevant to the pathogenesis of neurodegenerative diseases but that their interplay should be studied in greater depth.

Figure 2 still needs to be made easier to understand as it is very overloaded with text and images; a better way of displaying the information it contains must be found.

Author Response

Thank you for your thoughtful and detailed feedback on our manuscript. Your comments have been invaluable in helping us refine our work, and we appreciate your ongoing efforts to assist us in enhancing the quality of our research.

We agree with your suggestion regarding the conclusion. The intricate interplay between oxidative stress, gut microbiota, and epigenetic modifications warrants more in-depth exploration to fully understand their combined impact on the pathogenesis of neurodegenerative diseases. We have revised the conclusion to emphasize the need for further studies that focus on this complex interplay and its potential implications for developing novel therapeutic strategies. 

Regarding Figure 2, we acknowledge your concern about its complexity and difficulty understanding the information due to the overload of text and images. We have reduced the amount of text to ensure that the figure is more intuitive and easier to interpret.